

# Mineralogic controls on fault displacement-height relationships

Adam J. Cawood[1], David A. Ferrill[1], Kevin J. Smart[1], Michael J. Hartnett[1,a]

[1] Southwest Research Institute, 6220 Culebra Rd, San Antonio, Texas, USA

[a] present address: bonsAI LLC, San Antonio, Texas, USA

*Correspondence to*: Adam J. Cawood (adam.cawood@swri.org)

**Abstract.** Understanding the distribution and geometry of subsurface faults is critical for predicting fault penetration and associated leakage of fluids such as groundwater, hydrocarbons, and injected anthropogenic waste through sealing intervals. Fault dimensions are often underestimated due to the resolution limits of seismic reflection data, which only image portions of faults with sufficient displacement to offset seismic reflectors. To address this fault underestimation problem, we quantify

relationships between host rock composition and fault displacement gradients using a well-exposed outcrop of normal faults in mechanically layered sedimentary rocks in the footwall to the west branch of the Moab Fault, Utah. We integrate high-resolution digital photogrammetry, structural mapping, X-ray diffraction (XRD) mineralogy, and Schmidt rebound measurements to analyze how mineralogy and mechanical properties influence fault displacement vs. height relationships. Our results indicate that normal fault displacement gradients tend to be higher in less competent beds and lower in more competent

strata, and that fault displacement gradient is positively correlated with clay content and negatively correlated with strong minerals (e.g., quartz, feldspar, dolomite). Outcrop-derived relationships are used to build a predictive framework that uses fault displacement and mineralogy to predict fault height. We apply this framework to a worked seismic interpretation example and demonstrate that fault dimensions are likely substantially underestimated in conservative seismic interpretations. Our results highlight the importance of mechanical stratigraphy in controlling fault geometry and provide a data-driven approach

for estimating sub-seismic fault dimensions, with implications for reservoir characterization, fluid containment, and geohazard assessment.

## 1 Introduction

Although capable of acting as baffles or seals (e.g., Fossen et al., 2005; Childs et al., 2007), faults are widely recognized as conduits for subsurface fluid flow (Barton et al., 1995; Sibson and Scott, 1998; Faulkner et al., 2010; Roelofse et al., 2020;

Petrie et al., 2023), particularly in low-porosity and low-permeability rock (Caine et al., 1996; Evans et al., 1997; Gartrell et al., 2004; Ferrill and Morris, 2003; Ferrill et al., 2017a). As such, faults play a critical role in energy and resource systems, and fluid flow along faults is often beneficial for geothermal energy systems (e.g., Gan and Elsworth, 2014), aquifer recharge




and connectivity (e.g., Maclay and Small, 1983; Bauer et al., 2016), hydrocarbon migration (e.g., Allan, 1989; Fisher and Knipe, 2001), and the mobilization of mineralizing fluids that form or modify ore deposits (e.g., Garven, 1995; Cox, 2005).

Conversely, fault-controlled flow pathways can be detrimental for applications that rely on long-term fluid containment, such as hazardous waste disposal (e.g., Gautschi, 2001), greenhouse gas sequestration (e.g., Vialle et al., 2018), and hydrocarbon retention within subsurface traps.

A major uncertainty in evaluating the role of faults in subsurface systems is the difficulty of constraining their true dimensions. Fault continuity and dimensions strongly influence the potential for faults to breach sealing layers, connect rock volumes, and

intersect other faults and fractures, yet subsurface imaging consistently underestimates fault size. Seismic methods generally fail to image all but the largest faults in the subsurface (e.g., Marrett and Allmendinger, 1991; Yielding et al., 1996), and with the vertical resolution, or "limit of separability", of modern 3D broadband seismic data being typically on the order of ~10 meters (e.g., Duffy et al., 2015), faults with smaller displacements may remain entirely undetected. Larger faults are imaged only in segments where displacement exceeds the resolution threshold, leading to systematic underestimation of their true

vertical and lateral extents (Fig. 1). This limitation has critical implications for resource management and subsurface waste disposal, as interpretations may incorrectly suggest that low-permeability sealing intervals remain intact when they may, in fact, be compromised by undetected fault penetration.

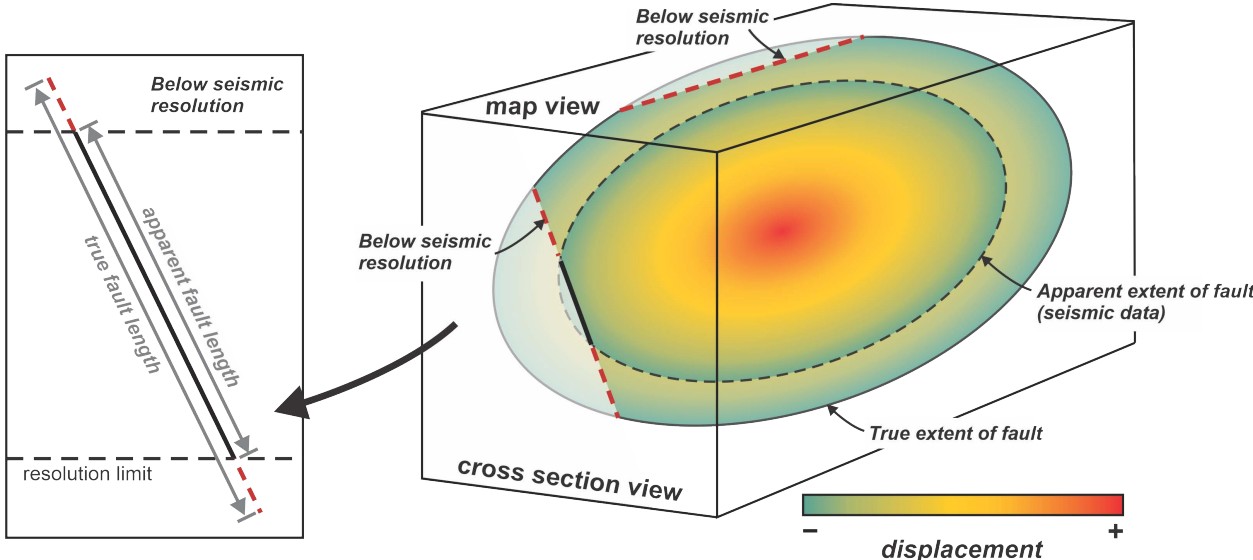

**Figure 1. Schematic representation showing the effect of seismic resolution on the apparent vertical and lateral extents of faults in**
**the subsurface. If sufficient displacement occurs along a fault (ca. 10-20 m, depending on data resolution), part of the fault may be observable in seismic reflection data. Variations in displacement on the fault surface and the resolution limits of seismic reflection data, however, will result in underestimation of fault dimensions.**

One approach to estimating sub-seismic fault dimensions is to use displacement-length scaling relationships (e.g., Scholz and Cowie, 1990; Clark and Cox, 1996; Kim and Sanderson, 2005; Torabi and Berg, 2011; Lathrop et al., 2022). While these




compilations capture general trends in fault displacement vs. length or height (Fig. 2), they also demonstrate substantial scatter in compiled data, making displacement alone an unreliable predictor of true fault dimensions in the subsurface. For example, a fault with ~100 m of displacement has a length or height that spans 3 orders of magnitude (~500 to 30,000 m). This variability has been attributed to a range of factors, including tectonic setting (e.g., Cowie and Scholz, 1992), fault kinematics and segment linkage (e.g., Peacock and Sanderson, 1991, 1996), and propagation or reactivation history (e.g., Kim et al., 2001). One of the most consistently observed controls on fault scaling is the compositional and mechanical properties of layered rocks. Mechanical stratigraphy has been shown to influence fault displacement vs. length or height in both extensional (e.g., Muraoka and Kamata 1983; Gross et al., 1997; Ferrill and Morris, 2003, 2008; Roche et al., 2012; Morris et al., 2014; Bowness et al., 2022) and contractional (e.g., Williams and Chapman, 1983; McConnell et al., 1997; Deng et al., 2013; Ferrill et al., 2016; Cawood and Bond, 2020) settings. A particularly important observation is that fault propagation tends to be inhibited in more ductile strata, producing higher displacement gradients within ductile units and lower gradients within more competent layers (Fig. 3; Muraoka and Kamata, 1983; Williams and Chapman, 1983; Ferrill et al., 2012, 2016; Cawood and Bond, 2020).

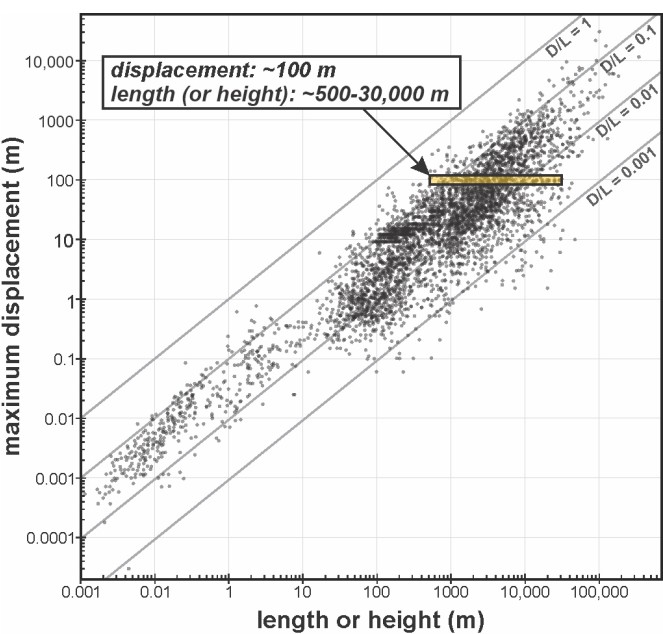

**Figure 2. Compilation of maximum fault displacement vs. fault length or height for normal faults. Modified from Lathrop et al. (2022). Yellow shaded box shows the range of potential fault lengths or heights for faults with displacements of ca. 100 m.**

Previous work on fault scaling has provided detailed measurements of fault length, height, and displacement, but lithological parameters are more often described qualitatively, using terms such as "semicompetent" or "less competent" (e.g, Muraoka and Kamata, 1983). Here we build upon previous work by integrating field observations and measurements, digital datasets, and laboratory analyses to quantitatively characterize relationships between mineralogy and fault displacement patterns at outcrop. Regression-based relationships between fault displacement gradient and XRD mineralogy from outcrop are then



applied to critically evaluate and revise a published subsurface fault interpretation. This approach uses outcrop exposures to quantitatively assess the influence of mechanical stratigraphy on deformation patterns and demonstrates that outcrop-based analyses can provide data-driven predictions of true fault dimensions in the subsurface, with direct implications for evaluating the integrity of sealing intervals in resource, waste, and storage systems.

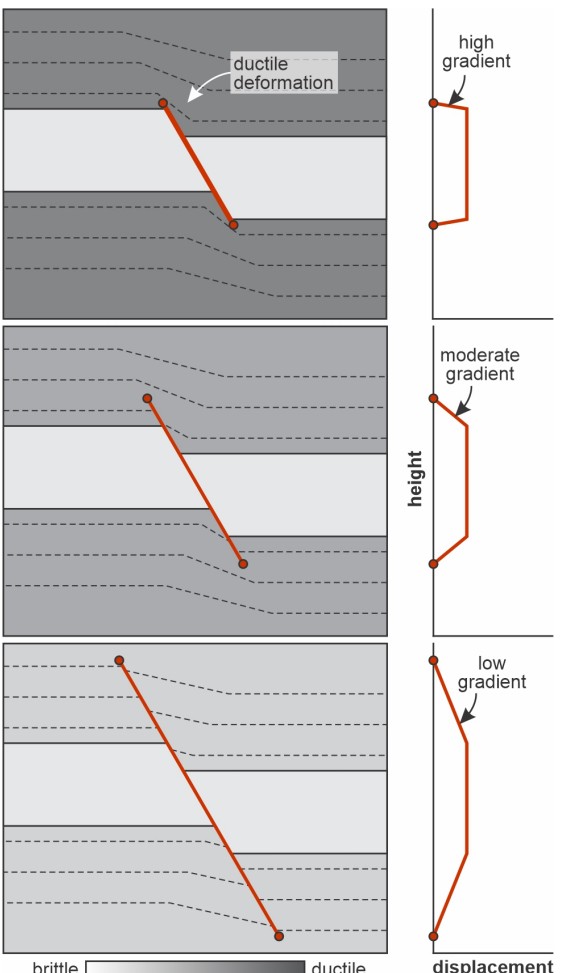


**Figure 3. Schematic diagram illustrating the effects of rock mechanical properties on fault height and displacement. Highly ductile units act as barriers to fault propagation, producing abrupt fault terminations and steep displacement gradients. More brittle lithologies exert less control, allowing faults to propagate more freely, resulting in lower displacement gradients.**



## 2 Study Area and Geologic Background

The Highway 191 roadcut exposure across from Arches National Park Visitor Center (Fig. 4a) lies approximately 8 km NW of Moab, Utah, and is a well-known site with exceptional normal fault exposures. The site was selected for this study for several reasons, including (1) exposure of over 100 normal faults within a mechanically layered succession, (2) clear lithologic boundaries between sedimentary layers, allowing fault cutoffs to be mapped confidently and precisely, (3) accessibility to the major units exposed at the site, which allowed us to collect samples and perform XRD analysis, and (4) exposure of mixed

siliciclastic and carbonate sedimentary rocks, including sandstones, siltstones, and mudrocks, which serve as important analogs for sandstone–mudrock reservoir–seal systems relevant to groundwater flow, hydrocarbon migration and trapping, and the containment of sequestered fluids and waste.

Structurally, the site is located in the footwall to the west branch or "railway splay" of the Moab fault, which juxtaposes the Moenkopi Formation in its hanging wall with the Honaker Trail Formation in its footwall at the study location (Fig. 4b, c;

Doelling, 1985; Foxford et al., 1998; Doelling et al., 2002; Ferrill et al., 2009). The west branch of the Moab fault has ca. 160 m of throw at the approximate position of the study site, compared with a maximum throw of approximately 960 m for the main Moab fault to the east (Foxford et al., 1998). The most recent slip on the Moab fault, along with associated fluid flow, has been dated to approximately 60–63 Ma based on $^{40}$Ar/$^{39}$Ar geochronology (Solum et al., 2005). Exposed at the site are a series of SW- and NE-dipping crossing conjugate normal faults with displacements of centimeters to meters that offset

sandstone, siltstone, and carbonate layers within the Pennsylvanian Honaker Trail Formation (Doelling et al., 2002; Ferrill et al., 2009). Slickenlines measured at the site by Ferrill et al. (2009) consistently indicate dip-slip displacement, with only slight obliquity of slip directions on some of the observed faults.

## 3 Data and Methods

### 3.1. Rock sampling, rebound data, and X-ray diffraction (XRD) analysis

A suite of 26 rock samples was collected from the study site for XRD mineralogy analysis. Sampling locations were selected to ensure that all sedimentary units identified in the field were represented at least once in the collected suite of samples. In some instances, multiple positions within the same bed were sampled. Where multiple samples were collected for the same bed, mean values for XRD mineralogy and rebound are reported and used for correlation analysis (Table 1). The full suite of sample data is provided in Table A1 (Appendix A). At each of the 26 sampling positions, an N-type Schmidt hammer was

used to measure in-situ elastic rebound, following the methodology of Morris et al. (2009). Rebound data were collected to evaluate the relative stiffness and strength of bedding layers, providing an estimate of rock competence. Although Schmidt hammer measurements do not directly measure rock stiffness or strength, they serve as a proxy for present-day rock mechanical properties (e.g., Young's modulus, unconfined compressive strength) in outcrop settings (Katz et al., 2000; Aydin and Basu,





2005). XRD compositional analysis of the collected samples was conducted by Ellington Geological Services, following the
procedures outlined in Bowness et al. (2022).

**Figure 4. Geologic setting and study location. (a) Oblique aerial image of the study site. View approximately towards the west. (b) Geologic map and (c) cross section showing regional geologic setting, modified from Doelling et al. (2002).Yellow stars in parts b and c show approximate study location.**



### 3.2. Digital photogrammetry

A total of 901 aerial images were captured at the study site for photogrammetric reconstruction. The images were taken using a 20-megapixel camera with a 24mm focal length, mounted on a DJI Phantom 4 Pro unoccupied aerial vehicle (UAV). Photos were acquired at fixed two-second intervals, with an ISO setting of 400, variable shutter speeds, and variable aperture values. Image collection followed established best practices (James and Robson, 2012; James et al., 2017; Cawood and Bond, 2018) to ensure sufficient overlap for successful photogrammetric processing. Photogrammetric reconstruction was performed using Agisoft Metashape Professional 1.7.3 (see Cawood et al., 2017 for details on processing steps). Image alignment and processing resulted in a cleaned and filtered dense point-cloud containing approximately six million points, with an average point spacing of 10.1 mm across an area of 6,070 m². From this dataset, a 3D photorealistic mesh with approximately one million faces was generated within Metashape Professional. The photogrammetric point-cloud and mesh were georeferenced using direct georeferencing, which integrates positional and orientation data recorded by the UAV's onboard Global Positioning System (GPS) and accelerometers. This approach eliminates the need for extensive ground control points while still providing approximate geospatial positioning, allowing the reconstructed model tied to real-world coordinates. We refer the reader to Nesbit et al. (2022) for a description and accuracy assessment of the direct georeferencing method.

### 3.3. Digital fault and horizon mapping, cross section construction, and displacement analysis

Faults and bedding horizons were interpreted in 3D using the polyline tool in Agisoft Metashape Professional following similar procedures to those described by Bowness et al. (2022). This approach allows precise mapping of structural features directly onto the photorealistic photogrammetric model. Fault and bedding horizon polyline interpretations were projected to a cross section oriented NE-SW (229°) in Move 2022.1TM (Petroleum Experts Ltd.), with the cross section orientation defined by the structural data of Ferrill et al. (2009), and a projection vector for polylines perpendicular to the cross-section. Projected polyline interpretations were resampled and adjusted where necessary to ensure consistency between fault and horizon interpretations 2D and 3D. Projected fault and horizon interpretations were used to measure fault throw, heave, and displacement in 2D to avoid measurement bias on non-planar surfaces in 3D.

Fault displacements were measured where faults cross mapped horizons in cross-section view, parallel to the mapped fault segment between measurement positions. Fault displacements were assumed to be within the plane of the constructed cross section, based on field observations of Ferrill et al. (2009) who documented that the majority of faults exposed at the site show dip-slip displacement, with only minor oblique slip on occasional faults. Where normal faults are interpreted to be offset by later thrust faults (Ferrill et al., 2009), this offset was restored prior to conducting displacement analysis on the normal faults.

### 3.4. Displacement gradient analysis and correlations

Fault displacement gradient (DG) was calculated using Eq. (1), where ΔD is the change in fault displacement between measurement positions, and h is the bedding-perpendicular distance between measurement positions:



$$DG = \Delta D/h \,, \tag{1}$$

Multiple displacement gradient values were calculated for several of the exposed beds at the site, where multiple analyzed faults cross the same mapped unit. Comparisons between displacement gradient, XRD mineralogy, and Schmidt rebound were performed using a Pearson correlation matrix for all variables and cross-plots of displacement gradient vs. (i) total clay (wt.
%), (ii) total carbonate (wt. %), (iii) the sum of quartz + feldspar + carbonate (wt. %), and (iv) Schmidt rebound (R) to capture mineralogical and mechanical influences on displacement gradient at the layer scale. For cross-plots, we fitted exponential ordinary least squares models by regressing displacement gradient on each of the variables described above.

### 3.5. Fault tip distance estimates

For a measured maximum displacement D on a mapped fault (e.g., from reflection seismic), the bed-perpendicular distance
from the displacement measurement to the either fault tip, T_dist   (upward or downward), is calculated following Eq. (2):

$$T\_dist = D/DG \,, \tag{2}$$

To predict T_dist  , we use outcrop-derived relationships between displacement gradient and bed-scale predictors (total clay, total carbonate, summed quartz + feldspar + carbonate, and Schmidt rebound) to estimate displacement gradient from rock properties via exponential fits (see Section 3.4 above). When D is measured at or near the maximum fault displacement, T_dist
approximates the half-height of the fault in section (total height ≈ 2 T_dist  ), assuming approximate symmetry. Uncertainties in predicted T_dist are estimated using 95% confidence intervals, allowing low- and high-case T_dist values to be calculated for a measured fault displacement and outcrop-derived estimate of displacement gradient. For prediction curves we use median displacement gradient per layer to limit the influence of outliers and local heterogeneity, and average XRD mineralogy and rebound to represent bed-scale composition and mechanical properties where multiple samples exist.

### 3.6. Seismic structural interpretations and application of fault tip distance predictors

We applied the outcrop-calibrated fault-tip distance method to a depth-converted seismic reflection profile from Cawood et al. (2022). Faults were first mapped conservatively (i.e., only where reflector offsets or terminations were unambiguous). We then generated predicted tip distances using: (i) the measured maximum displacement for each interpreted fault; (ii) an assumed clay content of 30 % for the stratigraphic section imaged in the seismic profile (following regional sand–shale ratios
summarized by Cawood et al., 2021); and (iii) the outcrop-derived relationships for displacement gradient vs. XRD mineralogy and rebound (Section 3.4), and associated estimates of distance to fault tip (Section 3.5). For each fault we computed median T_dist and low/high bounds by propagating 95% confidence intervals, and we applied the result upward and downward from the displacement maximum to estimate fault tip positions above and below the displacement measurement position. Fully revised, final fault interpretations were generated by manually refining the adjustments to fault interpretations described above.
Refinements included (i) occasional minor adjustments of predicted distances to fault tips based on seismic character and (ii)



explicit treatment of fault overlap vs. linkage where adjusted fault tips led to crossing or overlapping fault segments. The Highway 191 roadcut exposure across from Arches National Park Visitor Center (Fig. 4a) lies approximately 8 km NW of Moab, Utah, and is a well-known site with exceptional normal fault exposures. The site was selected for this study for several reasons, including (1) exposure of over 100 normal faults within a mechanically layered succession, (2) clear lithologic boundaries between sedimentary layers, allowing fault cutoffs to be mapped confidently and precisely, (3) accessibility to the major units exposed at the site, which allowed us to collect samples and perform XRD analysis, and (4) exposure of mixed siliciclastic and carbonate sedimentary rocks, including sandstones, siltstones, and mudrocks, which serve as important analogs for sandstone–mudrock reservoir–seal systems relevant to groundwater flow, hydrocarbon migration and trapping, and the containment of sequestered fluids and waste.

## 4. Results

### 4.1. Structural Interpretation

A total of 190 normal faults and 13 sedimentary horizons were identified and digitally mapped in 3D using the photogrammetric reconstruction of the study site (Fig. 5). Approximately 70% of the mapped faults dip towards the SW, with the remaining 30% dipping to the NE. We interpret these SW- and NE-dipping structures as a system of crossing conjugate normal faults, with NE-dipping and SW-dipping faults mutually cutting and offsetting each other (see Ferrill et al., 2009). Additionally, several low-angle thrust faults are exposed at the site. These thrust faults are offset by several of the mapped normal faults (e.g., Faults 3-5, Fig. 6), and conversely, several of the exposed normal faults are offset by the thrust faults (e.g., Faults 6-12, Fig. 6). This configuration suggests that extension (normal faulting) and contraction (thrust faulting) at the site were approximately coeval, with switching between extensional and contractional regimes (e.g., Ferrill et al., 2021). This switching of stress regime is consistent with a general interpretation for the site of normal fault development through outer arc extension (Ferrill et al., 2017b) above a contractional anticline formed by salt wall amplification during the Laramide Orogeny (Reeher et al., 2023).

### 4.2. XRD Mineralogy and Rebound

X-ray diffraction (XRD) analyses reveal substantial variability in mineralogical composition across the sampled stratigraphic units (Table 1). XRD and rebound data show marked variability across the units (Table 1). Quartz ranges from 18.5% (Unit J) to 64.3% (Unit C), and total clay from 7.5% (Unit M) to 38.2% (Unit D). Total feldspar (K-spar + plagioclase) spans 4.0% (Unit J) to 25.0% (Unit E). Total carbonate (calcite + dolomite) is lowest at ~1–1.3% (Units I and H) and highest at 67.5% (Unit J). The aggregate quartz + feldspars + carbonates ranges from 54.8% (Unit D) to 91.4% (Unit M). Quartz-rich examples include Unit C, Unit M, and Unit A (≥59–64% quartz with 8–18% clay). Clay-rich units include Unit D and Unit H (>35% clay). Carbonate-rich strata are less common, with Unit J standing out as an exceptionally out (>65% carbonate), with Units L–K at 23–30%. Schmidt rebound spans 5.5 (Unit D) to 60.3 (Unit C). Rebound values are generally higher in units with




higher quartz or carbonate content and lower in samples with elevated clay content. For example, quartz- or carbonate-rich

units (e.g., Units A, C, J, and M) exhibit higher rebound and clay-rich units (Units D and H) show lower rebound. XRD and

rebound data in Table 1 form the basis for the correlation analysis in the following sections (see Table A1 for the full suite of

data).

**Figure 5. (a) Digital outcrop model with interpreted stratigraphic horizons (white lines), labeled stratigraphic units (A–M, white text), and XRD sample locations (AR1–AR26, yellow boxes with black text). Sample locations AR16–AR26 lie outside the visible field of view and are shown in their projected stratigraphic positions. (b) Structural interpretation of the digital outcrop, showing 191 normal faults (black) and a thrust fault system (red). Fault terminations and intersections are color-coded to clarify interpreted fault relationships and geometries: white denotes observed fault tips, red marks fault–fault intersections, and blue indicates apparent fault termination at the edge of exposure. The model can be viewed and downloaded at https://sketchfab.com/3d-models/hwy-191-arches-roadcut-9876592de8a84c798b93bb5b263bc73e.**



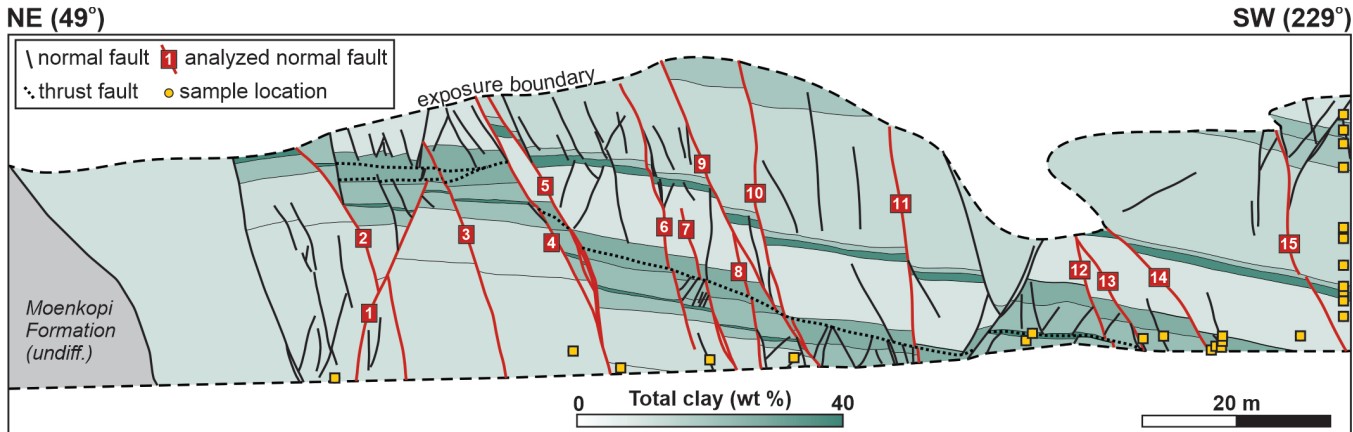

**Figure 6.** NE–SW cross section showing projected polyline interpretations of stratigraphic horizons, normal faults, and thrust faults. Sedimentary layers are colored by total clay content (%) derived from XRD mineralogy analysis. Normal faults analyzed in detail for displacement and displacement gradient are numbered and highlighted in red; see main text for details.

| Table 1. Summary of XRD mineralogy (reported in weight %) and rebound data | | | | | | | | | | | | |
|---|---|---|---|---|---|---|---|---|---|---|---|---|
| Unit | Rebound | Quartz | Potassium Feldspar | Plagioclase | Calcite | Dolomite | Pyrite | Anhydrite | Total Clay | Total Feldspars | Total Carbonate | Quartz + Feldspars + Carbonate |
| M | 51.2 | 63.6 | 7.1 | 3.2 | 2.6 | 14.8 | 0.5 | 0.3 | 7.5 | 10.4 | 17.4 | 91.4 |
| L | 37.6 | 42.8 | 8.8 | 4.5 | 0.5 | 23.1 | 0.7 | 0.2 | 18.8 | 13.3 | 23.6 | 79.7 |
| K | 35.9 | 40.3 | 8.9 | 3.2 | 3.6 | 26.5 | 1.0 | 0.1 | 16.1 | 12.1 | 30.0 | 82.4 |
| J | 46.9 | 18.5 | 2.7 | 1.3 | 35.3 | 32.3 | 0.5 | 1.1 | 8.2 | 4.0 | 67.5 | 90.0 |
| I | 40.5 | 50.6 | 11.3 | 9.8 | 0.3 | 0.7 | 1.2 | 0.3 | 25.3 | 21.1 | 1.0 | 72.7 |
| H | 17.1 | 40.5 | 7.8 | 8.4 | 0.2 | 1.0 | 4.6 | 0.3 | 36.5 | 16.2 | 1.3 | 58.0 |
| G | 38.9 | 51.1 | 9.0 | 7.8 | 1.1 | 11.8 | 1.9 | 0.2 | 16.7 | 16.8 | 12.9 | 80.8 |
| F | 41.9 | 44.5 | 13.0 | 6.9 | 0.9 | 4.5 | 5.3 | 0.3 | 24.3 | 19.9 | 5.4 | 69.8 |
| E | 17.0 | 49.2 | 14.7 | 10.3 | 3.9 | 0.9 | 1.4 | 0.3 | 18.8 | 25.0 | 4.8 | 79.0 |
| D | 5.5 | 36.6 | 8.4 | 7.2 | 1.8 | 0.8 | 6.2 | 0.2 | 38.2 | 15.6 | 2.6 | 54.8 |
| C | 60.3 | 64.3 | 8.6 | 5.3 | 1.0 | 1.4 | 1.1 | 0.3 | 17.7 | 13.9 | 2.4 | 80.6 |
| B | 46.0 | 56.0 | 8.3 | 5.5 | 1.3 | 7.9 | 2.3 | 0.2 | 18.2 | 13.8 | 9.2 | 79.0 |
| A | 60.2 | 59.4 | 8.0 | 3.4 | 1.9 | 18.0 | 0.6 | 0.2 | 8.1 | 11.4 | 20.0 | 90.7 |
| Rebound and XRD mineralogy data are reported as mean values where multiple samples were collected from a single unit. The full suite of data is provided in Appendix A. | | | | | | | | | | | | |



### 4.3. Normal fault displacements

Fault displacement analysis was performed for 15 normal faults at the study site (Fig. 6; Table A2). Although 190 faults were mapped across the outcrop, most were not suitable for quantitative analysis because they failed to meet necessary requirements. Faults were selected for displacement analysis based on the following criteria: (1) they offset mappable or clearly identifiable stratigraphic horizons, allowing displacement magnitudes and gradients to be calculated; (2) they are sufficiently large to offset multiple horizons, enabling multiple displacement measurements along individual faults; and (3) where possible, isolated or semi-isolated faults were chosen to minimize the influence of fault interaction, such as overlap or branching, which can locally distort displacement patterns. In cases where fault zones consist of multiple closely spaced segments, total (bulk) displacement across the zone was measured. As noted previously, several of the normal faults exposed at the site are offset by low-angle thrust faults – this contractional offset was restored on Faults 6-12 prior to measuring extensional displacement on normal faults.

Measured fault displacements range from zero at fault tips to a maximum of 7.08 m (Fault 2; Fig. 7). Of the 15 faults analyzed, only one fault (Fault 6) has both upper and lower tips exposed, five faults have a single exposed tip, and the remaining nine faults lack exposure of either tip. While several of the mapped faults at the site are exposed from tip to tip (Fig. 5), this tends to be more common for smaller structures that do not clearly offset multiple mapped horizons. Based on the selection criteria described above, these smaller faults were excluded from displacement analysis due to insufficient stratigraphic offset for reliable measurement. Fault displacement data for the 15 analyzed faults (Fig. 7) show that fault displacements vary substantially. Although no universal relationship between stratigraphic height and fault displacement is observed (i.e., a systematic increase or decrease in fault displacements in any given unit) there is some evidence that stratigraphic level influences patterns of fault displacement. For example, clay-rich units D and H are characterized by relatively abrupt changes in fault displacement. In contrast, fault displacements are relatively uniform through layers with lower clay content (e.g., units B and G). As noted above, these trends are not universal, however, and there are instances where clay-rich layers and clay-poor layers exhibit low and high displacement gradients, respectively.

### 4.4. Fault displacement gradients

Bulk relationships between composition, rebound, and displacement gradient were assessed by using a Pearson correlation matrix (Fig. 8). Pearson's r (−1 to +1) quantifies the strength and direction of a linear relationship, with the sign indicating direction and the magnitude indicating strength. Where a bed had more than one XRD or rebound sample (see Fig. 5), we averaged those values to a single bed-level estimate. For beds with multiple displacement gradient values, mean and median values were used for assessing correlations between displacement gradient, XRD mineralogy, and rebound. The full bed-by-bed displacement gradient dataset is provided in Table A2 (Appendix A). The correlation matrix indicates a consistent bed-scale compositional and mechanical control on fault displacement gradients. Mean and median displacement gradient show strong positive Pearson correlations with total clay (r = 0.88 and 0.93, respectively), strong negative correlations with Schmidt





rebound (r = 0.80 and 0.79), and moderate to strong negative correlations with calcite (r = 0.33 and 0.31), dolomite (r = 0.62 and 0.47), total carbonate (r = 0.58 and 0.47), and summed quartz, feldspars, and total carbonate (r = 0.87 and 0.87). These results suggest that clay-rich, lower-rebound beds tend to be associated with higher displacement gradients whereas beds with

higher Schmidt rebound and those beds dominated by stronger minerals (e.g., calcite and dolomite) are associated with lower displacement gradients. Note, correlations among individual minerals and aggregate sums (e.g., total carbonate) are largely driven by mineral co-dependence, and are therefore not interpreted. For coefficient values, we use "strong" to indicate r > 0.6, "moderate" for 0.3 < r < 0.6, and "weak" for r < 0.3.

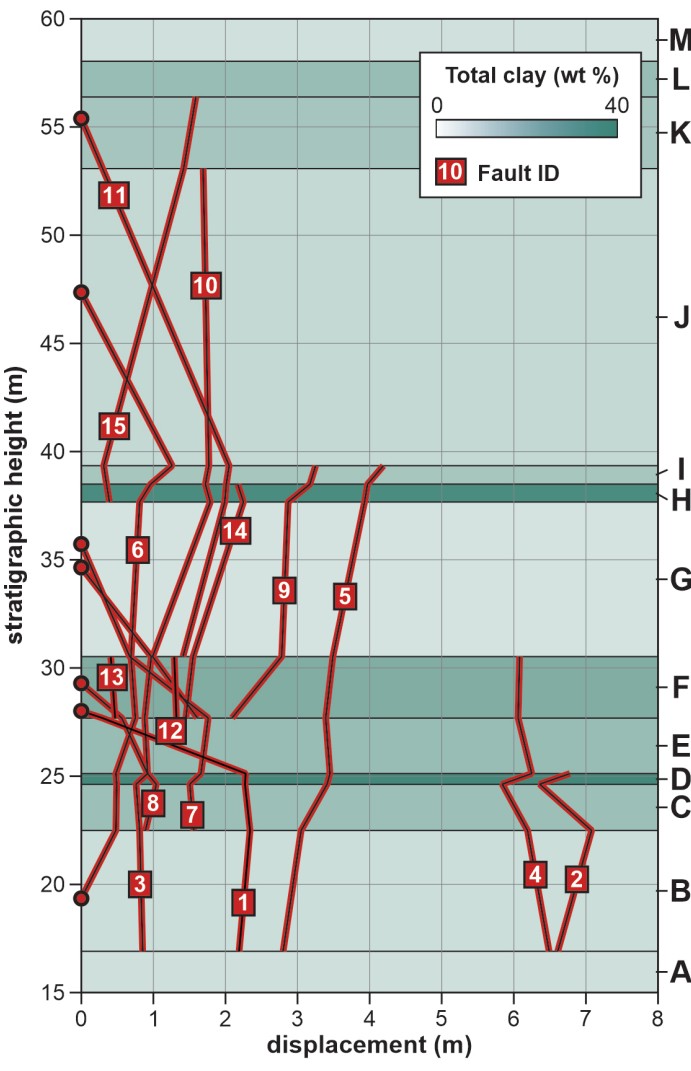

**Figure 7. Fault displacement vs. stratigraphic height for the 15 faults analyzed in detail. Stratigraphic interval colors correspond to clay weight percent from XRD analysis. Letters to the right of the plot denote assigned stratigraphic units (see Figure 5A). Note that faults 1-5 extend downwards past the base of interval A but the A-B boundary marks the lowermost position that displacements can be reliably measured. Red circles indicate fault tips that were observed in the outcrop exposure.**





Cross-plots of displacement gradient against mineralogy and rebound (Fig. 9a–d) reproduce the general patterns indicated by the correlation matrix (Fig. 8). Displacement gradient is positively correlated with total clay (Fig. 9a) and negatively correlated with (i) total carbonate (Fig. 9b), (ii) summed quartz, feldspars, and carbonate (Fig. 9c), and (iii) Schmidt rebound (Fig. 9d). The full measurement cloud (grey points) for each cross-plot shows generally consistent but noisy structure, with weak to very weak correlations ($R^2$ = 0.08-0.21). Layer medians (black bars) show median displacement gradient vs. average values for

XRD mineralogy and rebound for each mapped unit. We use median displacement gradient per layer to limit the influence of outliers and local heterogeneity, and average XRD mineralogy and rebound to represent bed-scale composition and mechanical properties where multiple samples exist.

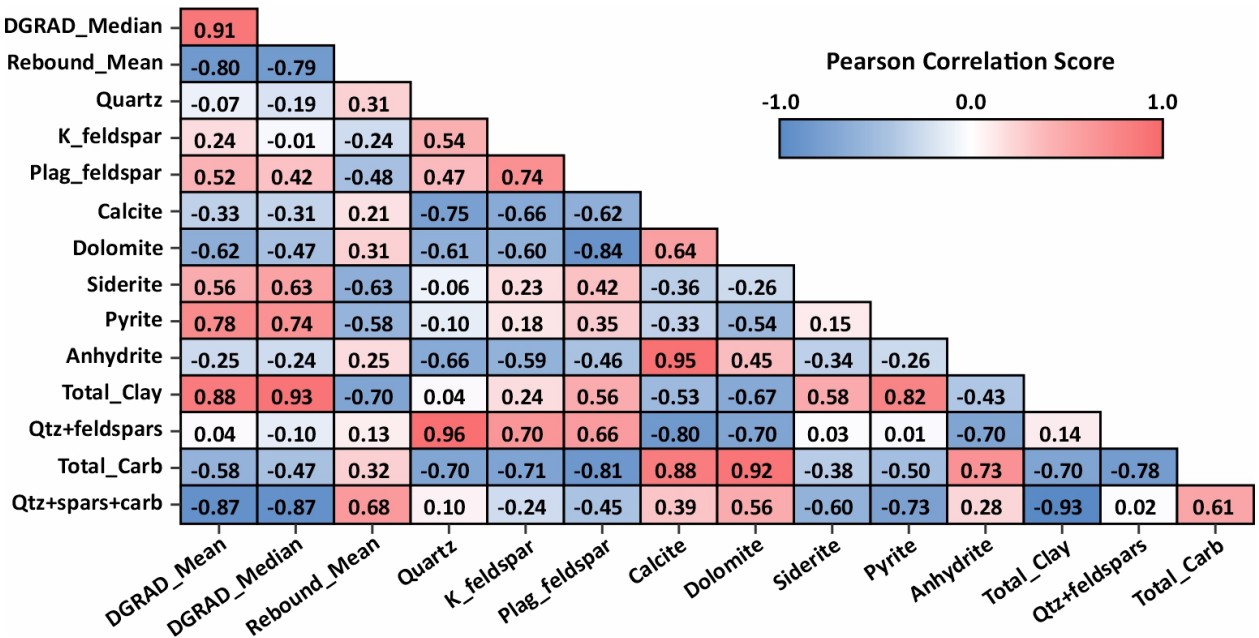

**Fig. 8. Pearson correlation matrix showing Pearson correlation coefficients between displacement gradient, XRD-derived**
**mineralogy, and Schmidt rebound values. Mean and median per-layer displacement gradients (DGRAD) show systematic correlations with mineralogic components: positive correlation with clay content, and negative correlations with stronger mineral components dolomite, quartz, and rebound. Correlations are also observed among mineralogical components themselves, likely reflecting co-dependence related to depositional processes.**

Relationships for median displacement gradient vs. mean mineralogy and rebound values show tighter trends on the cross-
plots and much stronger correlation coefficients in each case ($R^2$ = 0.28-0.95). Despite the differences observed in correlation coefficients for all points vs. bed averages in each cross-plot, exponential fits yield similar slopes and directionality, providing evidence that relationships between displacement gradient vs. mineralogy and Schmidt rebound are robust. Improved correlations for median displacement gradient vs. mean XRD mineralogy and rebound suggest that mineralogical controls on



displacement gradient are best expressed at the bed scale, whereas point-wise variability reflects local structure, exposure
limits, and measurement noise. While the Pearson correlation coefficients (r) reported in Fig. 8 differ from the  coefficients of
determination (R²) from exponential ordinary least squares fits reported in Fig. 9, they show the same general structure and
trends in the compiled data.

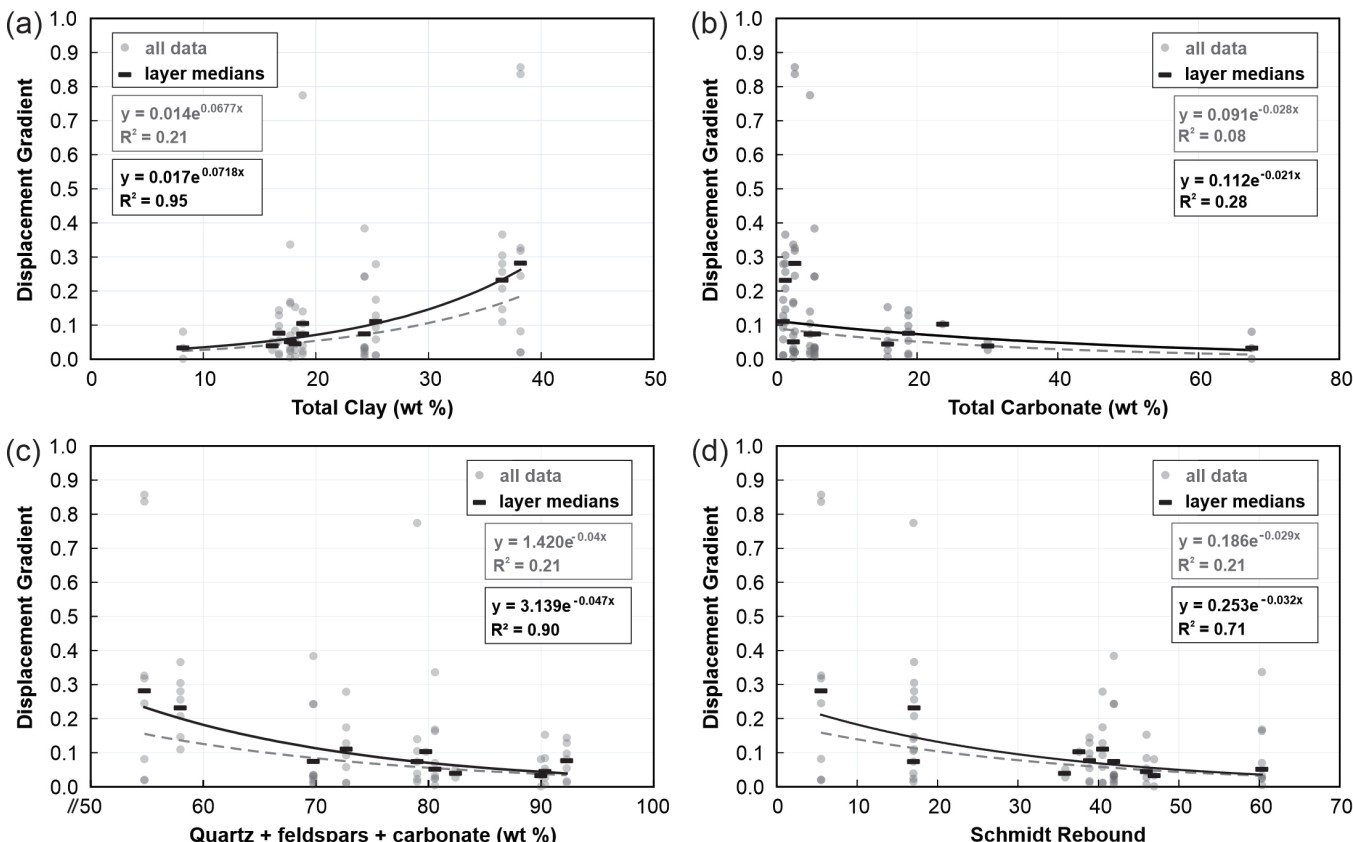

**Fig. 9. Cross-plots showing relationships between displacement gradient and (a) total clay content, (b) total carbonate content, (c)**
**the sum of quartz, feldspar, and carbonate, and (d) Schmidt rebound. Grey data points represent all individual displacement**
**gradient measurements plotted against corresponding mineralogy or rebound values. Black squares show layer-median**
**displacement gradient vs. mean values for mineralogy and rebound for each stratigraphic unit.**

**4.5. Predicted fault tip distances from fault displacements**

Based on observed relationships, modeled exponential fits, and correlations between fault displacement gradients, mineralogy,
and Schmidt rebound (Figs. 8 and 9), we generated a series of curves to predict layer-perpendicular distance to fault tip based
on displacement gradient (Fig. 10). A range of theoretical fault displacements, mineralogical compositions, and rebound values
are used so that, for example, a maximum measured fault displacement (e.g., 1 km) and host rock mineralogy (e.g., 30% clay)
can be used to predict the layer-perpendicular distance to the fault tip from the position at which fault displacement is measured.





Prediction curves are built using the layer averaged exponential equations in Fig. 9. The resulting families of tip-distance vs.
displacement curves (Fig. 10) show internally consistent behaviors across predictors. Increasing clay content is associated with
larger displacement gradients and, consequently, shorter distances to the fault tip for a given displacement (Fig. 10a). In
contrast, increases in total carbonate and in the summed fraction of quartz, feldspar, and carbonate correspond to lower
displacement gradients and thus longer distances to fault tips (Fig. 10b, c). The same pattern holds for Schmidt rebound, which
shows lower displacement gradients are associated with higher rebound values, and therefore longer tip distances for a given
displacement (Fig. 10d). In addition, we convert Schmidt rebound (R) to mechanical properties using the empirical
relationships of Katz et al. (2000) for rebound vs. Young's modulus E (in Gigapascals; GPa) and uniaxial compressive strength
U (in Megapascals; MPa). Specifically, we use Eq. (3) for Young's modulus:

$$\ln(E) \; = \; -8.967 + 3.091 * \ln(R) \,, \tag{3}$$

and Eq. (4) for uniaxial compressive strength:

$$\ln(U) \; = \; 0.792 + 0.067 * (R) \,, \tag{4}$$

Applied to our data, these transforms reproduce the rebound trends, showing that higher E and UCS correspond to lower
displacement gradients and therefore longer predicted distances to the fault tip (Figs. 10d and 10e).

### 4.6. Application of outcrop-derived relationships to a seismic structural interpretation

We apply our predicted displacement versus distance-to-fault-tip relationships to a worked example from the southern Salar
Basin, offshore Newfoundland, which experienced Late Jurassic and Cretaceous rifting associated with the opening of the
North Atlantic (see Cawood et al., 2022 geologic background and seismic reflection data). The uninterpreted seismic profile
shows a series of subhorizontal reflectors within a moderately extended stratigraphic section (Fig. 11a). Our conservative
interpretation of this seismic profile (Fig. 11b) represents an interpretation where faults were only mapped  where seismic
reflectors are clearly offset or truncated, with interpreted fault tips positioned where clear offset of reflectors transitions to
more ambiguous zones such as dipping reflectors or zones of opaque reflectors near the mapped fault trace. The conservative
interpretation yields 26 interpreted normal faults with maximum measured displacements of 14.2 to 110.4 m (Fig. 12; Table
2). This interpretation results in relatively short fault traces, and although there are indications of additional structures in the
seismic reflection profile, the absence of discrete reflector offset led us to exclude them from the interpretation (Fig. 11b).

The adapted version of the interpretation (Fig. 11c) incorporates maximum measured displacements on each fault, assumes a
uniform clay content of 30% for the entire stratigraphic sequence within the seismic profile, and uses our outcrop-derived
trends to predict fault dimensions. Due to the lack of wells in the area of the seismic profile (Cawood et al., 2022), a uniform
clay content of 30% was used based on regional sand–shale ratios, as  summarized in Cawood et al. (2021). Mid case (median





calculated Tdist) and high case (longer tip distances from 95% confidence envelope) tip positions are shown in Figure 11c,
which are defined by projecting mapped fault traces upward and downward to the predicted bed-perpendicular (vertical)
distance from the position where maximum fault displacement was measured. Low case tip position predictions are omitted
from Figure 11c for clarity.



**Figure 10. Fault tip distance curves derived from outcrop data for variable mineralogical and mechanical properties. Curves are based on displacement gradient trends observed in outcrop and show predicted fault tip distance as a function of measured fault**
**displacement (up to 500 m). Curves are generated from exponential fits in Figure 9 and shading shows 95% confidence intervals for the mean of each curve. Fault tip distance predictions are shown for variable (a) total clay content, (b) total carbonate content, (c) combined quartz, feldspar, and carbonate, (d) Schmidt rebound, (e) Young's modulus, and (f) uniaxial compressive strength, both of which are estimated using the best-fit equations of Katz et al. (2000). See main text for details.**





**Figure 11.** Application of fault displacement vs. fault tip distance predictions to a subsurface example from offshore Newfoundland. Seismic profile and fault interpretations modified from Cawood et al (2022). **(a)** Uninterpreted seismic profile. **(b)** Conservative fault interpretation, with fault traces only interpreted where seismic reflectors are clearly truncated and offset. **(c)** Adapted fault interpretation using the measured maximum displacement for each fault and predicted tip distances with an assumed clay content of 30% and outcrop-derived displacement gradient trends (see Fig. 10). White circles show originally interpreted fault tips in part b; blue circles show adjusted fault tip positions based on a mid-case scenario, and yellow circles show potential fault tip positions based on a high-case scenario (see main text and Table 2). Where adjusted interpretations result in overlapping faults (e.g., faults E, F, and G), traces are kept separate for clarity. **(d)** Final fault interpretation based on adjusted faults in part c. In some cases (e.g., faults E, F, and G), overlapping faults are joined in the final interpretation. In other cases (e.g., faults W and X), faults are interpreted as separate, overlapping structures.



Relative to the conservative fault interpretation, outcrop-derived fault tip-distance predictions yield longer fault lengths (increased bed-perpendicular distance to fault tips) for 58% of the mapped faults for low-case predictions (15 of 26 faults), 77% for mid-case predictions (20 of 26 faults), and 100% for high-case predictions (all 26 faults). For low case predictions, tip distance adjustments range from -173.3 m to +312.2 m, with a mean change of +21.3 m (Table 2). Tip distance adjustment factors (predicted tip distance divided by measured tip distance for conservative interpretation) for the low case range from

0.5 to 2.8, with a mean of 1.3. For the mid case, adjustments range from -87.2 m to +579.6 m (adjustment factors of 0.7-4.1), with a mean of +160.0 m (mean factor of 1.9). For the high case, all tip distances increase, from +6.2 m to +979.3 m (factors of 1.0–6.2), with a mean change of +366.3 m (mean factor of 2.8). The largest absolute increases in tip distances are for Fault F in the low case (+312.2 m) and for Fault P in the mid (+579.6 m) and high (+979.3 m) cases (Table 2). Fault tip adjustment factors > 1 indicate greater predicted fault heights than the original interpretation, whereas factors < 1 indicate smaller predicted

fault heights.

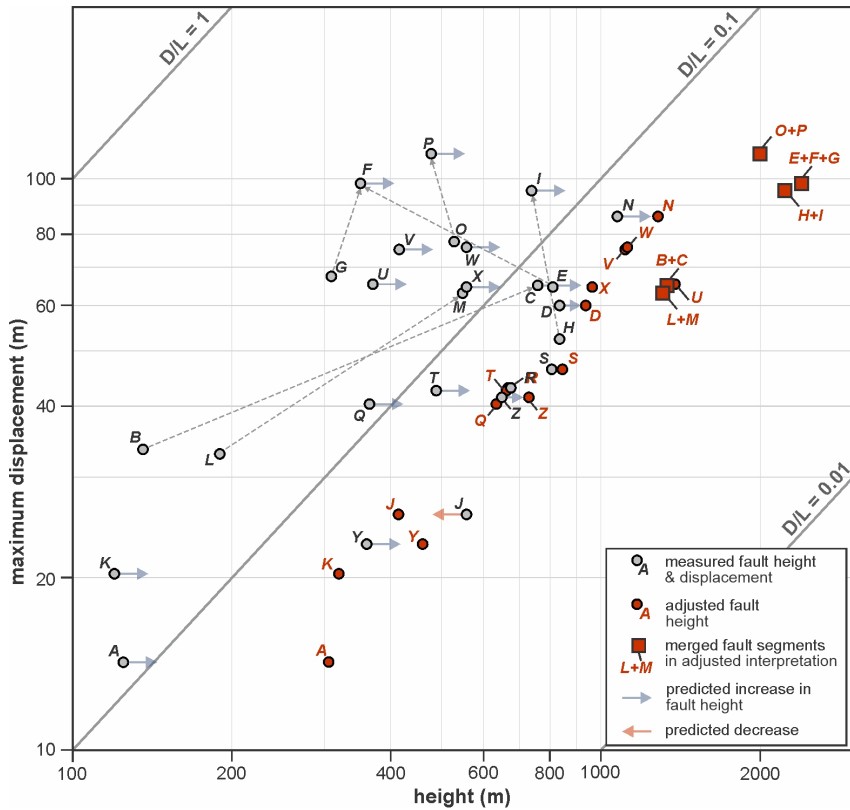

**Fig. 12. Maximum fault displacement vs. fault height for structural seismic interpretation example in Fig. 11. Dashed lines show faults that were merged and connected in the final, adjusted interpretation. Data for measured and adjusted fault heights are provided in Table 2.**






**Table 2. Measured displacements and measured/predicted fault tip distances and heights**

| Fault ID | Measured displacement (m) | Measured T_dist (m)* | Measured fault height | Predicted T_dist, low (m) | Predicted height, low (m) | T_dist adjustment, low (m) | Adjustment factor, high** | Predicted T_dist, mid (m) | Predicted height, mid (m) | T_dist adjustment, mid (m) | Adjustment factor, high** | Predicted T_dist, high (m) | Predicted height, high (m) | T_dist adjustment, high (m) | Adjustment factor, high** | Final fault heights (m) | Final Fault ID |
|---|---|---|---|---|---|---|---|---|---|---|---|---|---|---|---|---|---|
| A | 14.2 | 62.5 | 125.0 | 70.7 | 141.5 | 8.2 | 1.1 | 105.3 | 210.6 | 42.8 | 1.7 | 156.7 | 313.4 | 94.2 | 2.5 | 305 | A |
| B | 33.5 | 68.0 | 136.0 | 166.9 | 333.8 | 98.9 | 2.5 | 248.4 | 496.8 | 180.4 | 3.7 | 369.7 | 739.3 | 301.7 | 5.4 | -- | -- |
| C | 64.9 | 380.0 | 760.0 | 323.3 | 646.6 | -56.7 | 0.9 | 481.2 | 962.4 | 101.2 | 1.3 | 716.2 | 1432.3 | 336.2 | 1.9 | 1335 | B+C |
| D | 59.9 | 418.5 | 837.0 | 298.4 | 596.8 | -120.1 | 0.7 | 444.1 | 888.3 | 25.6 | 1.1 | 661.0 | 1322.0 | 242.5 | 1.6 | 937 | D |
| E | 64.5 | 406.0 | 812.0 | 321.3 | 642.7 | -84.7 | 0.8 | 478.2 | 956.5 | 72.2 | 1.2 | 711.8 | 1423.5 | 305.8 | 1.8 | -- | -- |
| F | 97.9 | 175.5 | 351.0 | 487.7 | 975.4 | 312.2 | 2.8 | 725.9 | 1451.8 | 550.4 | 4.1 | 1080.3 | 2160.6 | 904.8 | 6.2 | 2405 | E+F+G |
| G | 67.3 | 154.5 | 309.0 | 335.3 | 670.6 | 180.8 | 2.2 | 499.0 | 998.0 | 344.5 | 3.2 | 742.7 | 1485.3 | 588.2 | 4.8 | -- | -- |
| H | 52.3 | 418.0 | 836.0 | 260.6 | 521.1 | -157.5 | 0.6 | 387.8 | 775.6 | -30.2 | 0.9 | 577.1 | 1154.3 | 159.1 | 1.4 | -- | -- |
| I | 95.1 | 370.5 | 741.0 | 473.8 | 947.5 | 103.3 | 1.3 | 705.1 | 1410.2 | 334.6 | 1.9 | 1049.4 | 2098.8 | 678.9 | 2.8 | 2232 | H+I |
| J | 25.8 | 278.5 | 557.0 | 128.5 | 257.1 | -150.0 | 0.5 | 191.3 | 382.6 | -87.2 | 0.7 | 284.7 | 569.4 | 6.2 | 1.0 | 414 | J |
| K | 20.3 | 60.0 | 120.0 | 101.1 | 202.3 | 41.1 | 1.7 | 150.5 | 301.0 | 90.5 | 2.5 | 224.0 | 448.0 | 164.0 | 3.7 | 319 | K |
| L | 32.9 | 95.0 | 190.0 | 163.9 | 327.8 | 68.9 | 1.7 | 243.9 | 487.9 | 148.9 | 2.6 | 363.1 | 726.1 | 268.1 | 3.8 | -- | -- |
| M | 62.9 | 274.0 | 548.0 | 313.4 | 626.7 | 39.4 | 1.1 | 466.4 | 932.7 | 192.4 | 1.7 | 694.1 | 1388.2 | 420.1 | 2.5 | 1315 | L+M |
| N | 85.7 | 538.0 | 1076.0 | 426.9 | 853.9 | -111.1 | 0.8 | 635.4 | 1270.8 | 97.4 | 1.2 | 945.7 | 1891.4 | 407.7 | 1.8 | 1287 | N |
| O | 77.4 | 264.5 | 529.0 | 385.6 | 771.2 | 121.1 | 1.5 | 573.9 | 1147.8 | 309.4 | 2.2 | 854.1 | 1708.2 | 589.6 | 3.2 | -- | -- |
| P | 110.4 | 239.0 | 478.0 | 550.0 | 1100.0 | 311.0 | 2.3 | 818.6 | 1637.1 | 579.6 | 3.4 | 1218.3 | 2436.5 | 979.3 | 5.1 | 2005 | O+P |
| Q | 40.2 | 182.5 | 365.0 | 200.3 | 400.5 | 17.8 | 1.1 | 298.1 | 596.1 | 115.6 | 1.6 | 443.6 | 887.2 | 261.1 | 2.4 | 635 | Q |
| R | 43 | 338.0 | 676.0 | 214.2 | 428.4 | -123.8 | 0.6 | 318.8 | 637.6 | -19.2 | 0.9 | 474.5 | 949.0 | 136.5 | 1.4 | 674 | R |
| S | 46.3 | 404.0 | 808.0 | 230.7 | 461.3 | -173.3 | 0.6 | 343.3 | 686.6 | -60.7 | 0.8 | 510.9 | 1021.8 | 106.9 | 1.3 | 846 | S |
| T | 42.5 | 244.5 | 489.0 | 211.7 | 423.5 | -32.8 | 0.9 | 315.1 | 630.2 | 70.6 | 1.3 | 469.0 | 938.0 | 224.5 | 1.9 | 637 | T |
| U | 65.2 | 185.3 | 370.5 | 324.8 | 649.6 | 139.6 | 1.8 | 483.4 | 966.8 | 298.2 | 2.6 | 719.5 | 1439.0 | 534.2 | 3.9 | 1383 | U |
| V | 75 | 208.0 | 416.0 | 373.6 | 747.3 | 165.6 | 1.8 | 556.1 | 1112.2 | 348.1 | 2.7 | 827.6 | 1655.2 | 619.6 | 4.0 | 1118 | V |
| W | 75.8 | 278.5 | 557.0 | 377.6 | 755.2 | 99.1 | 1.4 | 562.0 | 1124.0 | 283.5 | 2.0 | 836.5 | 1672.9 | 558.0 | 3.0 | 1124 | W |
| X | 64.5 | 278.5 | 557.0 | 321.3 | 642.7 | 42.8 | 1.2 | 478.2 | 956.5 | 199.7 | 1.7 | 711.8 | 1423.5 | 433.3 | 2.6 | 965 | X |
| Y | 22.9 | 180.5 | 361.0 | 114.1 | 228.2 | -66.4 | 0.6 | 169.8 | 339.6 | -10.7 | 0.9 | 252.7 | 505.4 | 72.2 | 1.4 | 460 | Y |
| Z | 41.3 | 325.0 | 650.0 | 205.8 | 411.5 | -119.3 | 0.6 | 306.2 | 612.4 | -18.8 | 0.6 | 455.7 | 911.5 | 130.7 | 1.4 | 732 | Z |

*measured tip distances represent the average of distances upward and downward to fault tips from the point of maximum measured displacement

**Adjustment factors are calculated as predicted tip distance divided by measured tip distance in the conservative interpretation





In the conservative fault interpretation (Fig. 11b), all faults are mapped as isolated structures, with no overlap or intersection of structures. Adjustment of fault tip positions (Fig. 11c) results in overlapping or crossing fault geometries for 17 of the 26 (65%) mapped faults (using mid or high case fault tip distance adjustments). These overlapping or intersecting faults in the

adjusted interpretation (e.g., faults E, F, and G; Fig. 11c) highlight potential relay zones, fault splays, and linkages that require further examination and explicit treatment in the seismic interpretation. We performed a final stage of fault interpretation by manually refining and adjusting predicted fault tip positions. These adjustments were limited to (i) minor increases or decreases where seismic character and reflector continuity clearly support nearer or farther tips, and (ii) explicit handling of overlap zones. In some cases (e.g., faults E, F, and G), multiple overlapping traces were collapsed onto a single fault trace in the final

interpretation (Figs. 11 and 12), and in other cases, overlaps between structures were retained (e.g., faults W and X). Overall, the final interpretation preserves systematic increases in fault tip distances, as calculated from our outcrop-derived predictions (Figs. 11 and 12).

## 5. Discussion

### 5.1. Mineralogic and mechanical influences on fault displacement gradient

Our outcrop displacement measurements, XRD mineralogy, and Schmidt rebound data show that displacement gradients increase with clay content and lower rebound (weaker, more ductile beds), and decrease in high-rebound units dominated by stronger minerals (quartz, feldspar, dolomite, calcite). These results are consistent with previous work showing that fault propagation tends to be inhibited in more ductile clay-rich strata – where ductile deformation precedes brittle failure and inhibits fault propagation – producing higher displacement gradients within ductile units and lower gradients within more

competent layers (e.g., Muraoka and Kamata, 1983; Williams and Chapman, 1983; Ferrill and Morris, 2008; Ferrill et al., 2016; Cawood and Bond, 2020). As noted by Ferrill et al. (2017b), normal faults tend to nucleate in more competent clay-poor strata. Once a fault has nucleated, its propagation rate is largely set by the ductility of the host rock. Brittle, clay-poor units (e.g., massive limestone, indurated sandstone) allow tips to advance (propagate) rapidly relative to slip accumulation, producing low displacement gradients and little associated folding (e.g., Ferrill & Morris, 2008). As a result, for a given

displacement, faults in stronger, more brittle lithologies are expected to be larger (taller/longer) than faults in clay-rich, ductile sequences because fault displacement tends to decay less abruptly in more competent rock.

### 5.2. Limitations and future work

We acknowledge and expect that predictive relationships will vary with specifics of mineralogy and diagenesis, burial and deformation history, and deformation environmental conditions (including fluid pressure). Continued analysis and regional or

local calibration will be needed for application for different geological settings. The utility of our approach depends on several simplifying assumptions. For the purposes of this study, we assume uniform clay content and mechanical properties within





each stratigraphic unit, and apply empirical relationships derived from XRD mineralogy, Schmidt rebound measurements, and outcrop-scale deformation patterns. These assumptions allow us to isolate the influence of mineralogical composition on displacement gradient and fault tip behavior. The relationships we present are derived from a single well-exposed outcrop, and

therefore reflect the behavior of faults in one specific lithological and structural context. As such, they may not fully capture the range of fault scaling behaviors observed across different tectonic settings, burial histories, or mechanical stratigraphies. This represents a limitation in applying our model directly to other basins and settings without appropriate calibration. Further, we use a simple statistical model for predicting distance to fault tip from a limited dataset. Calibrated models would likely require larger datasets and more sophisticated statistical treatment of data that fully captures analytical (e.g., XRD precisions)

and measurement uncertainties.

Our predictive curves (Fig. 10) and case example (Figs. 11 and 12) assume fault propagation and growth through homogeneous media. Sedimentary sequences, however, are typically mechanically layered and heterogeneous. Increased layering or compositional contrast is likely to inhibit fault propagation and result in higher displacement gradients and shorter distances to fault tips (e.g., Ferrill and Morris, 2008; Morris et al., 2009; Ferrill et al., 2012). In such settings, fault propagation may be

arrested or inhibited at lithologic boundaries, and therefore our predicted fault tip distances (that assume homogenous media) may be somewhat higher than is appropriate for mechanically layered sequences in the subsurface. Additionally, the evolution of fault systems often involves complex interactions such as fault linkage, segment overlap, and displacement transfer between adjacent faults (e.g., Peacock, 1991; Bürgmann et al., 1994; Cartwright et al., 1995). These factors are not explicitly accounted for in our model but may substantially impact patterns of fault displacement and associated fault displacement gradients.

Similarly, the temporal evolution of host rock and fault mechanical properties may substantially alter patterns of fault displacement and the nature of fault zones at the sub-seismic scale. Strain hardening or softening, for example, may lead to temporal changes in fault slip vs. propagation ratios and an evolution of displacement gradients for a given fault zone.

Future work should aim to test and refine the displacement–tip distance relationships presented here by applying the framework to other outcrop analogs and across a broader range of lithologic and structural settings. Incorporating laboratory-derived

mechanical data, log-based mineralogy, and higher-resolution stratigraphic constraints would help to ground-truth predictions and quantify uncertainties. The integration of 3D seismic datasets could also allow for comparison between observed and predicted fault geometries at the basin scale, offering a means to validate or revise outcrop-calibrated trends. Together, these efforts would enhance the robustness and transferability of this framework and improve its utility for accurately predicting fault dimensions in the subsurface.

**5.3. Reducing uncertainties in subsurface fault interpretations**

Subsurface fault interpretations are inherently uncertain. Where subsurface data are sparse or poorly resolved (e.g., 2D rather than 3D seismic, depth-conversion uncertainty, variable image quality), fault dimensions may be difficult or impossible to



accurately estimate, and fault height or length may be highly uncertain (e.g., Dimmen et al., 2023). These uncertainties have important implications for fault penetration through sealing intervals, subsurface fluid flow and rock volume connectivity, and
subsurface risk assessments. Causes for these uncertainties include a range of factors, including (i) variable interpretation approaches (Michie et al., 2021), (ii) poor seismic imaging and low contrast/continuity that can broaden ranges of mapped fault geometries (Alcalde et al., 2017), and (iii) prior knowledge and mental models that may lead to drastically different interpretations of the same seismic profile (e.g., Bond, 2015).

In practice, interpreters may have substantially different styles of interpreting faults in seismic data, leading to substantially
different outcomes. One geoscientist may map a fault only where reflector offsets are unambiguous while another may extend a fault interpretation beyond clear visible evidence in seismic reflection data. Both fault interpretations and associated displacement–distance relationships may be consistent with global compilations (see Fig. 2), yet lead to different closures, seal assumptions, and risk outcomes. The result can be user-led mismatches across projects or business units, with consequences for volumetrics, well and infrastructure placement, seal/hazard assessment, and storage screening. Structural interpretations
are a major source of uncertainty for $CO_2$ sequestration projects (e.g., Carpentier et al., 2018; Osmond et al., 2020) and geothermal exploration (e.g., Diehl et al., 2017; Witter et al., 2019), and post-drill assessments have shown that trap and seal failure, and associated structural interpretations, are a common cause for dry oil and gas exploration wells (Knipe et al., 1997; Rudolph and Goulding, 2017; Murray et al., 2020). Robust fault interpretations are therefore of critical importance for resource appraisal, $CO_2$/hydrogen storage, groundwater protection, geothermal targeting, and hazard and infrastructure risk.

Our approach addresses the uncertainties described above by (i) focusing direct fault interpretation on high-confidence or relatively unambiguous faults, and (ii) providing a predictive framework that links measurable fault displacement and host rock mineralogy to expected fault tip distances, based on empirical trends observed in well-characterized outcrop analogs. By incorporating mineralogical controls on displacement gradient, this method enhances our ability to infer the true extent of faults, even in the absence of clear seismic indicators. In doing so, it offers a valuable alternative tool for refining fault models
and reducing geometric uncertainty in structural and reservoir characterization workflows. By making the links between composition, mechanical competence, and fault-tip distance explicit, and by bracketing plausible ranges, we narrow the space of admissible models and provide reasonable limits of fault height that can be carried forward into reservoir, seal, and hazard evaluations. This approach does not eliminate non-uniqueness, but it differentiates between aspects of fault interpretation that are relatively certain versus more interpretive, and makes the interpretive portion clearly visible, quantified, and tractable.

**6. Conclusions**

1. Our outcrop measurements show that fault displacement gradients are systematically related to host rock mineralogy and mechanical rock properties. We document higher displacement gradients in clay-rich units and lower gradients in rocks



dominated by stronger minerals such as quartz, feldspar, and dolomite. Displacement gradients also tend to be lower for units with higher Schmidt rebound values, reflecting the role of mechanical stiffness in controlling fault propagation and growth.

2. Our predictive framework linking displacement magnitude, host rock composition, and fault tip distance allows estimation of fault dimensions below seismic resolution in the subsurface. This approach is calibrated using outcrop data but leverages parameters (e.g., displacement, clay content) that are commonly available from subsurface datasets, including seismic interpretation, core analysis, and geophysical logs.

3. Application to a subsurface example from offshore Newfoundland demonstrates that conservative seismic interpretations

likely underestimate fault extent, particularly where reflector offsets are subtle or absent. This suggests that underestimation of fault dimensions may be widespread in seismic structural interpretations. This finding has broad implications for the reliability and robustness of analyses that rely on accurate fault interpretations.

4. Our framework reduces geometric uncertainty in structural and reservoir characterization by coupling rock composition and mechanical competence to fault dimensions. This approach places feasible bounds on fault dimensions for a given fault

displacement and host-rock composition, yielding defensible, robust estimates for reservoir, seal, and hazard evaluations.

**Appendix A: XRD mineralogy and displacement gradient data**

Summary XRD mineralogy and Schmidt rebound values are reported in the main text (Table 1). Table A1 shows the full dataset, including measurements for individual samples and the averaged values used to compute unit-level mineralogy and rebound. The fault displacement-gradient measurements underpinning the correlations and cross-plots in the main text (Figs.

8 and 9) are provided in Table A2.







**Table A1. Detailed XRD mineralogy (reported in weight %) and rebound data**

| Sample ID | Unit | Rebound (mean) | Quartz | Potassium Feldspar | Plagioclase Feldspar | Calcite | Dolomite | Pyrite | Anhydrite | Total Clay | Total Feldspars | Total Carbonate | Quartz + Feldspars + Carbonate |
|---|---|---|---|---|---|---|---|---|---|---|---|---|---|
| AR26 | M | 51.2 | 63.6 | 7.1 | 3.2 | 2.6 | 14.8 | 0.5 | 0.3 | 7.5 | 10.4 | 17.4 | 91.4 |
| AR25 | L | 37.6 | 42.8 | 8.8 | 4.5 | 0.5 | 23.1 | 0.7 | 0.2 | 18.8 | 13.3 | 23.6 | 79.7 |
| AR24 | K | 61.9 | 44.6 | 10.0 | 2.4 | 0.5 | 28.3 | 0.7 | 0.1 | 12.9 | 12.4 | 28.8 | 85.7 |
| AR23 | K | 9.8 | 36.0 | 7.7 | 4.0 | 6.6 | 24.7 | 1.2 | 0.1 | 19.3 | 11.8 | 31.3 | 79.1 |
| AR23+AR24 Mean | K | 35.9 | 40.3 | 8.9 | 3.2 | 3.6 | 26.5 | 1.0 | 0.1 | 16.1 | 12.1 | 30.0 | 82.4 |
| AR22 | J | 62.7 | 7.9 | 0.7 | 0.4 | 85.8 | 0.6 | 0.3 | 3.0 | 1.3 | 1.1 | 86.3 | 95.3 |
| AR21 | J | 11.3 | 30.9 | 4.0 | 2.2 | 5.6 | 44.0 | 0.7 | 0.2 | 12.1 | 6.2 | 49.6 | 86.7 |
| AR20 | J | 66.7 | 16.6 | 3.4 | 1.2 | 14.4 | 52.3 | 0.4 | 0.2 | 11.2 | 4.6 | 66.7 | 87.9 |
| AR20-AR22 Mean | J | 46.9 | 18.5 | 2.7 | 1.3 | 35.3 | 32.3 | 0.5 | 1.1 | 8.2 | 4.0 | 67.5 | 90.0 |
| AR19 | I | 39.2 | 41.3 | 11.7 | 9.4 | 0.4 | 0.8 | 0.3 | 0.3 | 35.0 | 21.1 | 1.2 | 63.7 |
| AR18 | I | 41.8 | 59.8 | 10.9 | 10.2 | 0.3 | 0.5 | 2.0 | 0.2 | 15.6 | 21.1 | 0.8 | 81.7 |
| AR18+AR19 Mean | I | 40.5 | 50.6 | 11.3 | 9.8 | 0.3 | 0.7 | 1.2 | 0.3 | 25.3 | 21.1 | 1.0 | 72.7 |
| AR17 | H | 17.1 | 40.5 | 7.8 | 8.4 | 0.2 | 1.0 | 4.6 | 0.3 | 36.5 | 16.2 | 1.3 | 58.0 |
| AR16 | G | 41.6 | 58.1 | 9.6 | 7.0 | 0.8 | 17.5 | 0.5 | 0.2 | 6.1 | 16.6 | 18.3 | 93.0 |
| AR15 | G | 58.0 | 54.8 | 9.7 | 7.9 | 2.2 | 16.9 | 0.5 | 0.1 | 7.5 | 17.6 | 19.2 | 91.6 |
| AR15+AR16 Mean | G | 38.9 | 51.1 | 9.0 | 7.8 | 1.1 | 11.8 | 1.9 | 0.2 | 16.7 | 16.8 | 12.9 | 80.8 |
| AR14 | F | 51.8 | 35.7 | 9.9 | 8.6 | 0.7 | 15.5 | 2.2 | 0.2 | 26.8 | 18.5 | 16.2 | 70.3 |
| AR13 | F | 45.6 | 44.2 | 13.1 | 5.2 | 1.6 | 2.3 | 6.4 | 0.4 | 26.5 | 18.3 | 3.9 | 66.4 |
| AR12 | F | 44.5 | 48.4 | 12.5 | 6.4 | 0.7 | 1.4 | 6.9 | 0.5 | 23.0 | 18.9 | 2.1 | 69.3 |
| AR11 | F | 39.9 | 45.9 | 10.0 | 7.3 | 0.9 | 2.1 | 7.7 | 0.2 | 25.6 | 17.4 | 3.0 | 66.3 |
| AR10 | F | 27.9 | 48.0 | 19.4 | 7.1 | 0.9 | 1.0 | 3.2 | 0.3 | 19.5 | 26.5 | 1.9 | 76.4 |
| AR10-AR14 Mean | F | 41.9 | 44.5 | 13.0 | 6.9 | 0.9 | 4.5 | 5.3 | 0.3 | 24.3 | 19.9 | 5.4 | 69.8 |
| AR9 | E | 25.6 | 52.6 | 14.9 | 9.7 | 3.6 | 0.9 | 1.2 | 0.3 | 16.1 | 24.7 | 4.6 | 81.8 |
| AR8 | E | 8.4 | 45.8 | 14.5 | 10.8 | 4.2 | 0.9 | 1.5 | 0.3 | 21.5 | 25.3 | 5.0 | 76.2 |
| AR8+AR9 Mean | E | 17.0 | 49.2 | 14.7 | 10.3 | 3.9 | 0.9 | 1.4 | 0.3 | 18.8 | 25.0 | 4.8 | 79.0 |
| AR7 | D | 5.5 | 36.6 | 8.4 | 7.2 | 1.8 | 0.8 | 6.2 | 0.2 | 38.2 | 15.6 | 2.6 | 54.8 |
| AR6 | C | 60.3 | 64.3 | 8.6 | 5.3 | 1.0 | 1.4 | 1.1 | 0.3 | 17.7 | 13.9 | 2.4 | 80.6 |
| AR5 | B | 55.4 | 65.0 | 8.3 | 3.4 | 1.2 | 14.7 | 0.7 | 0.1 | 6.3 | 11.7 | 15.9 | 92.6 |
| AR4 | B | 62.6 | 58.3 | 7.9 | 6.1 | 1.0 | 14.7 | 1.0 | 0.2 | 10.4 | 14.0 | 15.7 | 88.1 |
| AR4+AR5 Mean | B | 46.0 | 56.0 | 8.3 | 5.5 | 1.3 | 7.9 | 2.3 | 0.2 | 18.2 | 13.8 | 9.2 | 79.0 |
| AR3 | A | 64.7 | 68.1 | 12.1 | 2.7 | 0.5 | 11.4 | 0.5 | 0.2 | 4.3 | 14.9 | 11.8 | 94.8 |
| AR2 | A | 56.9 | 55.4 | 5.3 | 3.5 | 4.5 | 17.7 | 0.5 | 0.2 | 12.4 | 8.8 | 22.2 | 86.4 |
| AR1 | A | 59.1 | 54.7 | 6.6 | 3.9 | 0.8 | 25.1 | 0.9 | 0.1 | 7.6 | 10.5 | 25.9 | 91.0 |
| AR1-AR3 Mean | A | 60.2 | 59.4 | 8.0 | 3.4 | 1.9 | 18.0 | 0.6 | 0.2 | 8.1 | 11.4 | 20.0 | 90.7 |




**Table A2. Individual fault displacement measurements and associated displacement gradients for the study site**

| Fault ID | Unit | Measurement position height (m) | Displacement (m) | Displacement gradient |
|---|---|---|---|---|
| 1 | F | 30.25 | 0 | 0.11 |
| | E | 27.69 | 0.29 | 0.77 |
| | D | 25.12 | 2.28 | 0.02 |
| | C | 24.63 | 2.27 | 0.03 |
| | B | 22.49 | 2.34 | 0.03 |
| | A | 16.91 | 2.19 | |
| 2 | D | 25.12 | 6.78 | 0.86 |
| | C | 24.63 | 6.36 | 0.34 |
| | B | 22.49 | 7.08 | 0.08 |
| | A | 16.91 | 6.61 | |
| 3 | F | 29.29 | | |
| | E | 27.69 | 0.56 | 0.14 |
| | D | 25.12 | 0.92 | 0.33 |
| | C | 24.63 | 0.76 | 0.02 |
| | B | 22.49 | 0.81 | 0.01 |
| | A | 16.91 | 0.85 | |
| 4 | F | 30.53 | 6.08 | 0.01 |
| | E | 27.69 | 6.06 | 0.07 |
| | D | 25.12 | 6.25 | 0.84 |
| | C | 24.63 | 5.84 | 0.16 |
| | B | 22.49 | 6.19 | 0.05 |
| | A | 16.91 | 6.49 | |
| 5 | I | 39.35 | 4.16 | 0.13 |
| | H | 38.49 | 4.05 | 0.21 |
| | G | 37.67 | 3.88 | 0.05 |
| | F | 30.53 | 3.49 | 0.04 |
| | E | 27.69 | 3.39 | 0.02 |
| | D | 25.12 | 3.45 | 0.08 |
| | C | 24.63 | 3.41 | 0.17 |
| | B | 22.49 | 3.05 | 0.04 |
| | A | 16.91 | 2.8 | |
| 6 | J | 47.36 | 0 | |
| | I | 39.35 | 1.26 | 0.28 |
| | H | 38.49 | 1.02 | 0.26 |
| | G | 37.67 | 0.81 | 0.02 |
| | F | 30.53 | 0.69 | 0.02 |
| | E | 27.69 | 0.75 | 0.11 |
| | D | 25.12 | 0.48 | 0.02 |
| | C | 24.63 | 0.49 | 0 |
| | B | 22.49 | 0.48 | 0.15 |
| | A | 19.35 | 0 | |
| 7 | G | 35.73 | 0 | 0.13 |
| | F | 30.53 | 0.67 | 0.38 |
| | E | 27.69 | 1.76 | 0.04 |
| | D | 25.12 | 1.66 | 0.32 |
| | C | 24.63 | 1.5 | 0.03 |
| | B | 22.49 | 1.56 | |
| 8 | G | 34.65 | 0 | |
| | F | 30.53 | 0.97 | 0.03 |
| | E | 27.69 | 0.88 | 0.02 |



| | | | | |
|---|---|---|---|---|
| | D | 25.12 | 0.92 | 0.24 |
| | C | 24.63 | 1.04 | 0.07 |
| | B | 22.49 | 0.89 | |
| 9 | I | 39.35 | 3.25 | 0.09 |
| | H | 38.49 | 3.17 | 0.37 |
| | G | 37.67 | 2.87 | 0.01 |
| | F | 30.53 | 2.78 | 0.24 |
| | E | 27.69 | 2.09 | |
| 10 | J | 53.08 | 1.69 | |
| | I | 39.35 | 1.71 | 0.01 |
| | H | 38.49 | 1.72 | 0.28 |
| | G | 37.67 | 1.95 | 0.14 |
| | F | 30.53 | 0.92 | 0.24 |
| | E | 27.69 | 1.61 | |
| 11 | J | 55.39 | 0 | |
| | I | 39.35 | 2.05 | 0.17 |
| | H | 38.49 | 1.9 | 0.11 |
| | G | 37.67 | 1.99 | 0.08 |
| | F | 30.53 | 1.41 | |
| 12 | F | 30.53 | 1.28 | 0.01 |
| | E | 27.69 | 1.32 | |
| 13 | F | 30.53 | 0.39 | 0.03 |
| | E | 27.69 | 0.47 | |
| 14 | H | 38.49 | 2.5 | 0.3 |
| | G | 37.67 | 2.25 | 0.1 |
| | F | 30.53 | 1.55 | 0.04 |
| | E | 27.69 | 1.45 | |
| 15 | J | 53.08 | 1.42 | 0.08 |
| | I | 39.35 | 0.31 | 0.01 |
| | H | 38.49 | 0.3 | 0.15 |
| | G | 37.67 | 0.42 | |

## Author contributions

AC, DF, and KS performed data collection during the field work. AC collected UAV imagery, processed photogrammetry data, and performed all formal analysis for the paper. The draft manuscript and figures were prepared by AC and finalized after input and edits were provided by co-authors. Manuscript conceptualization by AC with input from co-authors through numerous discussions.

## Competing interests

The authors declare that they have no conflicts of interest.

## Code/Data availability

Data are provided in the manuscript and will be uploaded as supplemental information in the final version.



## Acknowledgements

Primary financial support for this work was provided by Southwest Research Institute Internal Research and Development project 15-R6297. Thanks to Ryan King at Ellington Geological Services for XRD analysis.

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
