# Peer review of "Mineralogic controls on fault displacement-height relationships"

_EGUsphere, 2025_

## Author Response (AR1)

The paper presents a methodology for predicting fault dimensions (specifically, fault height) from mineralogical (XRD) and mechanical (Schmidt Hammer) data in siliciclastic rocks. The study integrates structural and mineralogical datasets collected along a well-known cross-section of the west branch of the Moab Fault to develop statistical relationships between fault displacement gradient and XRD mineralogy. The Authors further apply the outcrop-derived relationship to a seismic interpretation of a kilometre-long section affected by normal faults from offshore Newfoundland.

I recommend minor revision. Some issues need to be addressed before manuscript publication in Solid Earth. I outline major and minor comments below.

**Accept. Thank you for the thorough and constructive review. Responses to individual comments are below.**

**Major comments:**

**Fault dimension parameters.** To understand the paper, it is fundamental the reader understands how the fault dimension parameters (i.e. displacement gradient and fault tip distance) are calculated. Although the (simple) calculations are clearly presented in the Methods section, I recommend including a graphical explanation of these parameters and equations by adding a second column to Figure 3. In this expanded figure, the Authors could illustrate an example of calculation of displacement gradient between two measurement points and an example of T_dist calculation. For T_dist, it is important to graphically show that the distance is calculated from the point where the measured displacement is maximum and that the total height corresponds to 2*T_dist. In this new figure, adjustment factors used in the seismic profile (lines 355-365, Table 2) could also be included.

**Accept. We have expanded Figure 3 (see parts c and d) to include a graphical illustration of displacement gradient between measurement points and definition of fault tip distance ($T$dist), including the relationship between $T$dist and total fault height (height ≈ 2·$T$dist when measured at maximum displacement). We have not included adjustment factors used in the seismic profile to keep our methods and results clearly distinct.**

**Lack of fault-tip exposure in the outcrop.** In my view, a limitation of the paper is that 9 faults out of 15 lack observable tips in the outcrop. Although the selection criteria stated by the Authors are reasonable (e.g., exclusion of smaller faults from the displacement analysis), the absence of exposed tips does not allow to verify the proposed relationship between fault tip distance and total clay content (Fig. 10a) undermining the applicability of the proposed method. The application of the prediction method to the seismic reflection profile is interesting, but subsurface can not replace outcrop validation.

**We agree that direct observation of fault tips provides the most straightforward constraint on fault extent; however, tip-to-tip exposure of faults is relatively**

uncommon in profile view, particularly for faults with more than a few meters of displacement, because full tip-to-tip fault height tends to exceed outcrop exposure dimensions. In contrast, fault tip measurements are more commonly obtained in map view, where lateral fault terminations may be exposed in plan-view pavements, particularly for smaller faults. As a result, the absence of exposed fault tips in cross-section reflects a fundamental geometric and scale-dependent sampling limitation rather than a shortcoming of the dataset or the proposed method.

The approach presented here is intentionally not intended to be calibrated exclusively using faults with exposed tips, nor does it require tip-to-tip exposure for application. Instead, it is explicitly designed to operate under realistic outcrop and subsurface conditions where fault tips are rarely observable, particularly in cross-sectional view. We have clarified in the revised manuscript that the relationships presented are intended to provide predictive constraints on fault extent rather than direct outcrop-scale validation in all cases, and that the seismic example illustrates application of the workflow rather than a substitute for outcrop validation.

**Large scatter in displacement-gradient data.** In Figures 10a and 10c (core business of the manuscript), the displacement gradients exhibit a wide range of values for any given mineralogical composition (e.g., for ~25% total clay, displacement gradients span 0.01–0.4), resulting in very weak correlations. The Authors have used median displacement gradients to improve correlations and reduce the influence of outliers and local heterogeneity (line 275-276). However, the scatter appears to reflect more than just heterogeneities or outliers and may instead relate to additional factors influencing fault propagation in mechanically layered media. These factors are briefly listed in the Discussion (lines 411–422), but I recommend expanding this section in light of the presented data.

We agree that the wide range of displacement-gradient values observed for a given mineralogical composition likely reflects more than local heterogeneity or statistical outliers. We have revised the Discussion to more explicitly acknowledge that displacement gradients in mechanically layered media are influenced by multiple interacting factors in addition to bulk mineralogy, including layer thickness, mechanical layering architecture, fault maturity, displacement magnitude, and proximity to fault tips or linkage zones.

Median displacement gradients are used to capture first-order trends and reduce sensitivity to local variability; however, we now emphasize that the observed scatter represents a physically meaningful range of fault behaviors rather than noise. The revised Discussion expands on these additional controls and clarifies that mineralogy provides a statistically robust but non-unique constraint on displacement gradient, consistent with the natural complexity of fault propagation in layered rocks.

**Section 5.3 can be shortened.** Conversely, the first part of section 5.3 of the Discussion (lines 431-449) can be shortened or removed since it adds little to the manuscript in its current form.

**Accept. We have substantially shortened Section 5.3.**

**Uniform clay content assumption.** In the final part of the Discussion (lines 423-429), it is important the Authors more fully address the implications of assuming a uniform clay content throughout the entire stratigraphic sequence in the seismic profile.

**Accept. Thank you for this comment. We have revised the final paragraph of the Discussion to more explicitly address the implications of applying a single effective clay content in the seismic example. The revised text clarifies that incorporation of log-based mineralogy and higher-resolution stratigraphic constraints would allow displacement gradients to be assigned on a unit-by-unit basis and improve quantification of uncertainty where strong vertical or lateral mineralogical variability is present.**

**Minor comments:**

Delete lines 176-184 since they are identical to section 2.

**Accept. Text deleted.**

Sample location is square, not rounded, please update the legend accordingly.

**Accept. Figure 6 legend updated.**

Move the text in lines 226-235 in section 3, as this explanation pertains to the Methods rather than Results.

**Accept. Text moved to section 3.**

Do the same for lines 310-315, which should be relocated to Section 3.1.

**Accept. Text moved to section 3.1.**

**Anonymous Referee #2**

The manuscript presents a well-structured and data-rich study that aims to improve fault dimension estimates below seismic resolution by correlating displacement gradients with host rock mineralogy. The authors use UAV-based photogrammetry, digital fault mapping, XRD mineralogy, and rebound measurements to build a predictive framework. While the paper is well structured and clearly written, and the integration of outcrop observations with seismic interpretation is a notable strength, the manuscript is highly technical and narrowly focused. It assumes a substantial level of prior familiarity with photogrammetry, seismic workflows, and fault modeling techniques, which may limit its accessibility to a broader geoscience readership. That being said, the study presents important implications and has strong potential to improve our understanding of fault-zone architecture and its representation across scales.

Below, I provide several comments aimed at improving the content and enhancing the manuscript's readability and impact for a wider scientific audience.

**Accept. We thank the reviewer for their constructive review of the manuscript. We agree that the study is necessarily technical, reflecting the complexity of the datasets and methods involved. In response to this comment, we have made several targeted revisions aimed at improving accessibility and clarity for a broader geoscience audience. Specifically, we have expanded Figure 3 to include a graphical explanation of key fault dimension parameters (displacement gradient, fault tip distance, and fault height), clarified terminology more consistently throughout the text, and added brief explanatory context in the Geological Background and Discussion to guide interpretation of the Results. We expect these revisions to improve readability and conceptual clarity while retaining the level of detail required for reproducibility and application by specialists.**

**Major Comments:**

**1-** The manuscript relies heavily on one well-exposed outcrop near Moab, Utah. While the mechanical layering and fault patterns are compelling, the general applicability to other tectonic settings (e.g., compressional, strike-slip, or different lithologies like basalt or evaporites) is not sufficiently discussed. Authors might consider including a discussion section (or expand 5.2) with a critical evaluation of where and how this predictive framework may fail or require recalibration.

**Accept. We have expanded Section 5.2 to explicitly discuss the conditions under which the framework may require recalibration or may be less applicable, including contractional and strike-slip fault systems, and lithologies with markedly different mechanical behavior (e.g., evaporites or crystalline rocks). The revised text clarifies that the approach is intended as a transferable workflow rather than a universal scaling law, and that application to other settings requires local calibration of displacement gradient relationships.**

**2-** Diagenesis, cementation, and fluid-rock interaction can alter rock mechanical properties and mineralogy post-deposition, yet this is only lightly touched upon. Authors can expand the discussion briefly to discuss how diagenetic overprint could alter XRD or Schmidt rebound values, affecting the model's input assumptions.

**Accept – thank you for this comment. Text added to address the potential for diagenetic effects on XRD and Schmidt rebound values**.

**3-** The manuscript contains repetitive sections that significantly affect clarity and readability. In particular, identical paragraphs appear multiple times (e.g., lines 80–87 and 177–184 are repeated verbatim). In addition, the distinction between "fault tip distance" and "fault height" is not always clearly maintained in the text. Although fault height is derived from twice the fault tip distance, the manuscript at times moves between these terms without explicitly restating their relationship, which may cause confusion for readers less familiar with fault-scaling terminology. Clarifying this distinction more consistently throughout the manuscript would improve readability.

**Accept. This point about identical paragraphs was also raised by Reviewer 1 and we have addressed it by revising the manuscript accordingly (this was an author error when transferring text to the submission template). We have also expanded Figure 3 to provide a clearer graphical explanation of fault dimension parameters and the relationship between fault height and fault tip distance.**

**Line-specific comments:**

**L56:** Here it could be beneficial to explain briefly what is the mechanical stratigraphy as the readers from the broad disciplines may not be familiar with the term.

**Accept. Text added for explanation of mechanical stratigraphy.**

**L65:** Here, the manuscript claims that lithological parameters in previous fault scaling studies are "more often described qualitatively," and only cites Muraoka and Kamata (1983) to support this statement. This significantly underrepresents the breadth of fault scaling and mechanical stratigraphy research over the past several decades. The authors should either revise this statement to be more nuanced or support it with a broader and more up-to-date set of references (e.g., Lathrop et al. 2022, Frontiers in Earth Sciences). As written, it reads as dismissive and does not reflect the maturity of the field.

**Accept. Thank you for this comment. We have removed this sentence from the text. The preceding paragraphs provide an overview of previous work on this topic (including Lathrop et al., 2022 and other recent studies) and as such we have not duplicated this material here.**

**L79:** The "Study Area and Geological Background" section is too brief to provide adequate geological context for the faults being analyzed. While the manuscript later mentions that some of the normal faults were reactivated as thrusts, the authors do not clearly explain the tectonic evolution that led to this inversion, or the deformation phases responsible for fault formation and reactivation. A short paragraph can be added to outline the tectonic history of the studied basin and surrounding region, the timing and style of extensional and compressional phases and whether the studied faults are part of a known inversion system. This addition would significantly strengthen the manuscript by grounding the outcrop observations in a well-established tectonic framework.

**Accept. We have revised the Geological Background to explicitly state that the normal faults studied here were not reactivated during later contraction. We now clarify that crosscutting relationships between normal and thrust faults described in the Results reflect superposition of distinct fault sets, not inversion or reactivation of normal or thrust faults. We have also added further geologic background to ground field observations within a broader tectonic framework.**

**Line 110–115:** (XRD Analysis): Clarify which specific minerals were used in cross-plots and how "total clay" was defined. Was smectite differentiated from illite or chlorite for example?

**Accept. Additional detail provided.**

**L141:** Again, the manuscript refers to inversion-related deformation (e.g., reactivation of normal faults as thrusts) without having previously introduced the concept in the geological background. This makes it difficult for the reader to fully understand the structural evolution of the fault network.

**Accept. See earlier response. We have revised the Geological Background to explicitly state that the normal faults studied here were not reactivated during later contraction. And have clarified that crosscutting relationships between normal and thrust faults reflect superposition of distinct fault sets, not inversion or reactivation of individual faults. We have also added further geologic background that we hope will help to ground field observations within a broader tectonic framework.**

**L153:** The approach for estimating fault tip distance using measured maximum displacement and an outcrop-derived displacement gradient is interesting and appears central to the predictive framework. However, it is not clear whether this approach is entirely novel or derived from previously established fault scaling methods. I recommend the authors explicitly clarify whether Equation (2) and the predictive workflow are: Based on existing published methods (in which case a citation is needed), or A new approach developed in this study (in which case, that novelty should be stated clearly).

**We have revised the text to explicitly acknowledge that the conceptual basis for using displacement–distance relationships and displacement gradients to infer fault tip positions was established by Williams and Chapman (1983). We clarify that Equation (4) (previously Eq. 2) builds on this prior work by formalizing the relationship between measured displacement and displacement gradient into a simple, explicit expression that enables direct prediction of fault tip distance and fault height. The revised text makes clear that the approach represents a modified and operationalized application of established concepts, rather than a wholly new approach.**

**L177:** The paragraph between lines 177 and 184 is identical to the one already presented earlier in the manuscript (Lines 80–87). This should be corrected.

**Accept. Text between lines 177 and 184 deleted.**